# A Simple Laser-Induced Breakdown Spectroscopy Method for Quantification and Classification of Edible Sea Salts Assisted by Surface-Hydrophilicity-Enhanced Silicon Wafer Substrates

**DOI:** 10.3390/s23229280

**Published:** 2023-11-20

**Authors:** Han-Bum Choi, Seung-Hyun Moon, Hyang Kim, Nagaraju Guthikonda, Kyung-Sik Ham, Song-Hee Han, Sang-Ho Nam, Yong-Hoon Lee

**Affiliations:** 1Department of Chemistry, Mokpo National University, Jeonnam, Muan-gun 58554, Republic of Korea; namubum@naver.com (H.-B.C.); tmdgus8020@naver.com (S.-H.M.); 2Plasma Spectroscopy Analysis Center, Mokpo National University, Jeonnam, Muan-gun 58554, Republic of Korea; hyang@mokpo.ac.kr (H.K.); naga.gk@mnu.ac.kr (N.G.); 3Department of Food Engineering, Mokpo National University, Jeonnam, Muan-gun 58554, Republic of Korea; ksham@mokpo.ac.kr; 4Division of Navigation Science, Mokpo National Maritime University, Jeonnam, Mokpo-si 58628, Republic of Korea; hansh@mmu.ac.kr

**Keywords:** edible salts, laser-induced breakdown spectroscopy, elemental analysis, classification

## Abstract

Salt, one of the most commonly consumed food additives worldwide, is produced in many countries. The chemical composition of edible salts is essential information for quality assessment and origin distinction. In this work, a simple laser-induced breakdown spectroscopy instrument was assembled with a diode-pumped solid-state laser and a miniature spectrometer. Its performances in analyzing Mg and Ca in six popular edible sea salts consumed in South Korea and classification of the products were investigated. Each salt was dissolved in water and a tiny amount of the solution was dropped and dried on the hydrophilicity-enhanced silicon wafer substrate, providing homogeneous distribution of salt crystals. Strong Mg II and Ca II emissions were chosen for both quantification and classification. Calibration curves could be constructed with limits-of-detection of 87 mg/kg for Mg and 45 mg/kg for Ca. Also, the Mg II and Ca II emission peak intensities were used in a *k*-nearest neighbors model providing 98.6% classification accuracy. In both quantification and classification, intensity normalization using a Na I emission line as a reference signal was effective. A concept of interclass distance was introduced, and the increase in the classification accuracy due to the intensity normalization was rationalized based on it. Our methodology will be useful for analyzing major mineral nutrients in various food materials in liquid phase or soluble in water, including salts.

## 1. Introduction

Salt is mainly made up of ionic compounds such as chlorides and sulfates of alkali and alkaline earth metals [1]. As an additive or a raw material, it is widely used for various foods. However, salt is a mineral extracted from several natural sources, which are seawater, underground rock, brine wells, and salt deserts. While the main component of salt is NaCl, it also contains other elements such as Mg, Ca, Sr, K, Li, Al, Si, Ti, Fe, S, O, H, and so on [2,3,4,5,6]. These minor elements show significant variations in their concentrations depending on their sources and production methods [6]. This variation in concentration of minor elements makes salts have different tastes and also potential health benefits or risks [2,7,8,9,10,11,12,13]. Therefore, quantification of minor elements in edible salts is essential for quality control in the salt production industry. Also, it should be noted that simple compositional profiles, not the quantitatively determined concentrations, of minor elements contained in edible salts can be utilized as chemical fingerprints for discriminating edible salt products according to their origins. Isotope ratio analysis has been generally accepted as a dependable methodology for discriminating minerals, food materials, and natural products so far [14,15,16]. Certainly, it would be a valuable scientific challenge to develop a methodology discriminating edible salt products based on isotope ratio analysis. However, for practical applications, simple elemental analysis techniques would be enough for discrimination of edible salts rather than the complicated isotope ratio analysis. The fact that the concentrations of minor elements in edible salts depend on their geographical origins and production methods allows the simpler elemental analysis techniques to classify edible salt products with dependable accuracy.

Laser-induced breakdown spectroscopy (LIBS) is one of the elemental analysis techniques based on optical emission spectroscopy [17,18,19]. In typical LIBS analysis, a pulsed laser beam is focused on the solid sample’s surface. The concentrated laser pulse energy ablates and ionizes atoms in and around the laser focal spot to ignite a plasma. The laser-induced plasma releases energy in the forms of light, sound, and heat in a few tens of microseconds. The optical emission from the laser-induced plasma is utilized for both quantification and material classification. In comparison with well-developed conventional elemental analysis techniques such as inductively coupled plasma optical emission spectroscopy (ICP-OES), inductively coupled plasma mass spectrometry (ICP-MS), and atomic absorption spectroscopy (AAS), LIBS has a few advantages such as simplicity in sample preparation, rapidness in analysis, and the capability of performing analysis in air that make it much simpler in its instrumentation. Despite these merits, LIBS is known to be inferior to ICP-OES, ICP-MS, and AAS in terms of limit of detection (LOD), precision, and accuracy. Fortunately, high performances in elemental analysis are not required to analyze minor elements in edible salts such as Mg, Ca, and K. The detection capability of alkali and alkaline earth metals at the concentrations of several hundred parts per million and higher was found to be enough [20]. LIBS is particularly strong at detecting alkali and alkaline earth metals, which are important in salt quality control and product discrimination. Typical nanosecond LIBS using a high-power flash-lamp-pumped Q-switched laser and a high- or medium-resolution spectrometer coupled with an intensified charge-coupled device (ICCD) camera detector can detect Mg, Ca, and K at the level of a few parts per million [21]. This implies that there is still more room to simplify not only the LIBS instrument but also the sample preparation for the purpose of salt analysis.

In this work, a simple salt analysis methodology based on LIBS was developed, and its quantification and classification performances were investigated using six popular edible sea salt products consumed in South Korea. The LIBS instrument was assembled with a low-power, compact diode-pumped solid-state laser (DPSSL) and a low-resolution miniature spectrometer. Also, the salt samples were prepared using laser-patterned silicon wafer (LPSW) substrates. Salt is actually a chemically inhomogeneous mixture of various ionic compounds and also comes in forms of powder with crystals of different sizes. For the sake of obtaining reliable results from LIBS analysis, such an inhomogeneous powder sample is typically milled, mixed, and pressed into a pellet. This sample preparation process forms a homogenized solid pellet. This pelletization process requires heavy equipment, namely a ball mill and a hydraulic press. However, salt is highly soluble in water. Thus, a large amount of salt powders can be sampled and dissolved in water. Sampling only a small part of the solution is enough to obtain a representative result because the solution is homogeneous. For LIBS analysis, the sampled solution is dropped onto a solid substrate and dried. When the solution is dried on the silicon wafer without laser patterning, the residual salt crystals are deposited in a very confined area. This makes the repetition of precise measurement hardly possible. Thus, it is necessary to spread out the residual crystals over a pre-defined area. Drying the solution on the LPSW substrate, the residual crystals can be spread over a patterned area. Then, multiple measurements can be performed on it [22]. Directly measuring salt powders would greatly simplify the LIBS analysis of salts by removing the complicated sample preparation process. However, there are issues to be resolved in powder LIBS. One of them is the blown-off particles forming a stationary cloud during successive laser ablations. There is a recent study by Rajavelu et al. resolving this issue [23]. Due to the limitations on wavelength coverage and detection capability of the low-cost LIBS instrument used in this work, only three elements, Na, Mg, and Ca, could be detected with confident identification. However, the emission line intensities of Mg and Ca observed in the LIBS spectra showed strong correlations to the corresponding concentrations determined using ICP-OES. This formed dependable calibration curves converting the measured emission intensities into the corresponding concentrations. To develop a classification model, the emission peak intensities of Mg II at 279 nm and Ca II at 393 nm were also employed as latent variables. The classification accuracy was investigated using the *k*-nearest neighbors (*k*-NN) algorithm [24,25]. When the intensities of the Mg II and Ca II lines were normalized by the emission line intensity of the Na I at 589 nm, the classification correctness could be significantly improved to 98.6%. This was rationalized using the concept of “increase in interclass distance”. Our results indicate that the low-performance LIBS instrument, and the simple sample preparation process using LPSW substates can be utilized as not only a practical quality assessment technique on the salt production sites but also an origin determination methodology in combination with an appropriate modelling algorithm.

## 2. Materials and Methods

### 2.1. Salt Samples and ICP-OES Analysis

For the six salt products used in this work, their codes, origins, types, and the concentrations of Mg and Ca are listed in Table 1. The concentrations of Mg and Ca in the sea salt samples were determined via ICP-OES. First, 15 g of each salt sample was dissolved in 85 g of ultrapure water (resistivity = 18.2 MΩ·cm) first. The 15 wt% salt solution was then diluted to the final factor of 1/5000. Mixtures of NaCl and MgSO_4_ were prepared to obtain powders in which the concentrations of Mg were 103.01, 206.50, 1014.2, 2027.4, 10,097, and 30,015 mg/kg. Also, mixtures of NaCl and CaCl_2_ were prepared to form powders with 99.171, 198.92, 497.29, 993.88, 2484.7, and 4971.9 mg/kg of Ca. These two sets of binary mixtures were diluted by a factor of 1/5000 using ultrapure water for the calibration standards for Mg and Ca. The calibration standards and sample solutions were fed into an ICP-OES spectrometer (GREEN, SPECTRO, Kleve, Germany) at the rate of 1 L/min. A power of 1400 W was applied to generate the plasma. Argon gas was used for coolant, auxiliary, and nebulizer (cross) flows at rates of 12.0, 1.0, and 1.0 L/min, respectively. The calibration curves for Mg and Ca were obtained using the Mg II and Ca II lines at 279.553 nm and 396.847 nm, respectively.

### 2.2. LIBS Analysis

The preparation process of LPSW substrates is as follows. Briefly, a silicon wafer (2-inch diameter) was placed on a motorized sample stage (L-406.40DD10, L-406.40DD10, and L-306; Physik Instrumente GmbH & Co. KG, Karlsruhe, Germany), and a lattice pattern was engraved on it over a 1 cm × 1 cm area using a DPSSL (1064 nm, 7 ns pulse duration, 270 μJ pulse energy, DTL-329QT, Laser-export Co. Ltd., Moscow, Russia). The DPSSL was operated at a repetition rate of 1 kHz, and the stage was moved at a rate of 1 mm/s. The DPSSL beam was focused through an objective lens (ten times magnification, 20 mm focal length). The pattern was composed of a set of forty 1 cm long horizontal trenches and the other set perpendicular to the horizontal ones. The space between two adjacent trenches was 250 μm, and the width and depth of each trench were approximately 55 and 28 μm, respectively. Pieces of plastic tape were attached along the four sides of the pattern. The four pattens were carved on each 2-inch-diameter silicon wafer. A total of twelve patterns were prepared using three silicon wafers. Two patterns were used for each salt sample. Figure 1a shows the four patterned areas on a silicon wafer. A 15 μL droplet of 15 wt% salt solution was dropped on the prepared pattern. The laser-produced trenches enhanced the surface hydrophilicity significantly, and the hydrophobic plastic tape confined the salt solution to the 1 cm × 1 cm area precisely. After being dried in an oven at 60 °C for 1 h, salt crystals were formed and distributed over the patterned area. Figure 1b shows a microscope image of the salt crystals formed on the patterned area.

A picture of the laboratory-assembled LIBS setup used in this work is shown in Figure 2. The same DPSSL, focusing optics, and sample stage were used as those for preparing LPSW substrates. A continuous-wave diode laser beam (wavelength = 635 nm, PL202, Thorlabs, Inc., Newton, NJ, USA) was sent to the sample surface through a pinhole and a variable neutral-density (ND) filter (NDC-25C-4, optical density = 0.04 − 4.0, Thorlabs, Inc., Newton, NJ, USA) to keep the lens-to-sample distance constant. When the lens-to-sample distance is changed, the diode laser spot image shifts on the detector plane of the vision camera (1.6 megapixels, color CMOS camera, CS165CU/M, Thorlabs, Inc., Newton, NJ, USA) through the short-pass filter (cut-off wavelength = 850 nm, FES0850, Thorlabs, Inc., Newton, NJ, USA). The lens-to-sample distance was adjusted to obtain the hottest plasmas first. Then, the position of the diode laser spot imaged by the vision camera was carefully monitored through all the measurements. However, the samples used in this work were prepared using flat silicon wafers with constant thicknesses. This allowed us to keep conducting measurements without re-adjusting sample surface height after the lens-to-sample distance had been optimized first. Optical emissions from ignited plasmas were collected using two plano-convex lenses (7-cm focal length, 2-inch diameter) and sent to a non-gated miniature spectrometer (MAYA2000PRO, Ocean Insight, Inc., Orlando, FL, USA) through an optical fiber (600 μm core diameter). The spectrometer covered the wavelengths from 200 to 650 nm with a resolution of ~1 nm. For recording LIBS spectra, the laser repetition rate was set to 1 kHz, and the sample placed on the motorized stage was translated at the rate of 1 mm/s. Each LIBS spectrum was taken from a 9.5 mm long line scan over the laser-patterned area. This launched 9500 laser pulses along the 9.5 mm long line. The detector exposure time was set to 10 ms, and 950 successively detected signals were averaged to obtain a line scan. This made the detector stay open through the whole 9.5 mm line scan, and it was not gated with any delay from a laser pulse. Thus, the continuum background emission, particularly strong from early plasmas, could not be removed from our LIBS spectra. Two 1 cm × 1 cm patterned areas were used to analyze each salt solution. On each patterned area, 31 parallel line scans were conducted. Thus, 61 (=31 line scans × 2 patterned areas) spectra were recorded for each salt sample. Considering the length (9.5 mm) and width (55 μm) of ablated trenches, the coverage of the 31 line scans was roughly estimated to be 16.2 mm^2^ of the 100 mm^2^ pattern area. Assuming that the salt crystals were spread evenly on the patterned area, the laser ablation sampled ~16% of the whole salt crystals.

## 3. Results and Discussion

### 3.1. Quantification of Mg and Ca

Figure 3a shows the representative LIBS spectra of the samples A, C1, C2, K1, K2, and V in the wavelength region between 200 nm and 650 nm. The three emission peaks of Mg, Ca, and Na employed for the following analysis are highlighted by a yellow-colored background. Intensity values for the following analysis were taken as baseline-subtracted peak areas, which is represented in Figure 3b. The intensity is the value corresponding to the red-colored area. First, we added up the intensity values across the Mg II emission peak, which correspond to the sum of the red- and blue-colored areas. Then, the baseline function was taken as the line connecting the two left- and right-most spectral intensity data. The area under this straight baseline (blue-colored area) was subtracted from the sum of the red- and blue-colored areas. The highlighted peaks are the strongest emissions of the corresponding elements. This approach would make up for low performance of the LIBS instrument used in this work. The Mg emission peak at 280 nm is composed of several close-lying Mg II lines at 279.078, 279.553, 279.800, and 280.270 nm which were not resolved due to the limited resolution of the miniature spectrometer used in this work. The Ca emission peak at 393 nm is a single Ca II line. The Na emission peak at 589 nm is the unresolved sodium D-line doublet at 589.0 and 589.6 nm. While the Na emission intensities, *I*_Na_, remain almost the same across the six salt samples, the intensities of Mg and Ca emissions, *I*_Mg_ and *I*_Ca_, respectively, vary significantly from one sample to another. Na is one of the matrix elements composing NaCl. Thus, *I*_Na_ was selected as a reference signal for intensity normalization. In addition to *I*_Na_, the performance of the total emission intensity (sum of those from 200 nm to 650 nm), *I*_Tot_, as a reference signal was also investigated. The performances of *I*_Na_ and *I*_Tot_ in improving the measurement precision were evaluated by comparing the relative standard deviations (RSDs) of *I*_Mg_, *I*_Mg_/*I*_Tot_, and *I*_Mg_/*I*_Na_ (Figure 4a) and also those of *I*_Ca_, *I*_Ca_/*I*_Tot_, and *I*_Ca_/*I*_Na_ (Figure 4b). In the cases of both the Mg and Ca emission peaks, the intensity normalization was found to be effective at decreasing the RSDs, that is, improving the measurement precision. However, between the two reference signals, *I*_Tot_ and *I*_Na_, the latter showed better performance. The average RSDs of *I*_Mg_/*I*_Na_ over the six samples is 15.2%, which is much smaller than those of *I*_Mg_ (23.1%) and *I*_Mg_/*I*_Tot_ (19.3%). Also, the average RSDs of *I*_Ca_/*I*_Na_ (23.1%) taken for the six samples were consistently decreased from those of *I*_Ca_ (34.2%) and *I*_Ca_/*I*_Tot_ (30.7%). In consideration of possible intensity saturation of the strongest Na I emission at 589 nm, the weaker Na emissions identified at 330 nm and 569 nm can also be used as reference signals for intensity normalization (see the assignments in Figure 3a). However, the weaker Na emissions showed much larger RSDs in their intensities than those of the strongest one. Thus, the intensity normalization using the weaker ones as references was not effective. In our experiment, the measurement precision was affected by two factors: (i) instrument performances and (ii) sample preparation. The LIBS instrument is inexpensive but provides low laser power (270 μJ/pulse) and low spectral resolution (~1 nm). The low laser power may result in relatively large plasma temperature fluctuations, leading to low measurement precision. Also, the salt crystals may not be spread evenly on the patterned area.

Calibration curves converting *I*_Mg_, *I*_Mg_/*I*_Tot_, *I*_Mg_/*I*_Na_, *I*_Ca_, *I*_Ca_/*I*_Tot_, and *I*_Ca_/*I*_Na_ into the concentrations of the corresponding analytes are shown in Figure 5. The concentrations of Mg and Ca, listed in Table 1, were determined for the six salt samples via ICP-OES. The concentration of Mg ranges from 0 mg/kg to 22,650 mg/kg. Although the concentrations of Ca vary with samples, the difference between the largest (2092 mg/kg) and smallest (134.2 mg/kg) ones is not as large as that of the Mg concentrations. The calibration curves were obtained by fitting the experimental values using two linear functions, *y* = *a* + *bx*, and *y* = *bx*. In fitting experimental values using the former function, the offset was floated to be determined. However, the determined offsets were smaller than the corresponding uncertainties in all of the fits (see the fitted parameter, *a*, noted in each panel of Figure 5). Thus, it would be reasonable to fit the experimental values with the offsets fixed to 0 using the latter function, *y* = *bx*. Also, the coefficients of determination, *R*^2^, are closer to 1 when the experimental values were fitted by *y* = *bx* than *y* = *a* + *bx*. The *R*^2^ values from the two fits are noted in each panel of Figure 5. Taking the ratios, *I*_Mg_/*I*_Na_ and *I*_Ca_/*I*_Na_, might lead to nonlinear calibration functions because the Na content decreases as the Mg or Ca contents increase. The nonlinear behavior would be observed clearly if the correlation between *y* (*I*_Mg_/*I*_Na_ or *I*_Ca_/*I*_Na_) and *x* (Mg or Ca concentration) was investigated over wide enough concentration ranges and/or *y* values were measured with high precision. In addition to these factors, self-absorption effects on the used analyte and reference emission peaks can play an important role in determining the form of calibration functions. In our experiment, the measured intensity ratios could be correlated with the concentration of Mg or Ca simply with linear functions, not nonlinear functions. If the samples contain much higher contents of Mg and Ca, nonlinear functions should be considered as calibration curves.

LODs of Mg and Ca in this method were estimated using the following equation [19]:(1)$$LOD=\frac{3\sigma }{b}$$

In the above equation, *σ* is the standard deviation of the measured quantity corresponding to y in the fitting functions. To estimate LODs, it is reasonable to take *σ* from *y* measured for the standard containing minimum amount of the analyte. *b* is the slope of linear fitting function, that is, sensitivity. As can be seen in Figure 5, fitting the experimental values with the offset fixed to 0 (*y* = *bx*) gave slightly higher sensitivity. In consideration of precision (*σ*) and sensitivity (*b*), LODs of Mg and Ca were estimated from fitting *I*_Mg_/*I*_Na_ and *I*_Ca_/*I*_Na_ with the linear function of *y* = *bx* and obtained to be 87 mg/kg and 45 mg/kg, respectively. As mentioned above, only a tiny volume (15 μL) of salt solution (15 wt%) was used for the analysis. Considering this sample volume used for the analysis, the absolute LODs of Mg and Ca were estimated to be 8.1 × 10^−9^ mol and 2.5 × 10^−9^ mol, respectively. These LOD performances of our low-power, low-resolution LIBS instrument are still sufficient for the analysis of Mg and Ca in edible salt products containing a few hundred to percent levels of Mg and Ca.

In addition to precision and LODs, calibration accuracy was also evaluated for the quantification of Mg and Ca. To express the accuracy quantitatively, root-mean-square errors (RMSEs) of predicted concentrations were calculated using the following equation:(2)$$RMSE=\sqrt{\frac{\sum_{i=1}^{n}{\left(C_{i}^{pred}-C_{i}^{ref}\right)}^{2}}{n}}$$

In this equation, *n* and *i* represent the total number of test data and the index given to each of them, respectively. $C_{i}^{pred}$ and $C_{i}^{ref}$ are predicted and reference concentrations corresponding to the *i*th test data. Herein, the leave-one-sample-out cross-validation (LOSO-CV) method was used to calculate RMSE values [20]. In LOSO-CV, one of the six data points for each calibration curve shown in Figure 5 was selected as the test data, and the calibration curve was constructed by fitting the other five data. The calibration function was then used to predict the concentration of the data left for testing. This process was repeated six times with different test data selected in turn. The LOSO-CV did not use any information of the sample corresponding to the selected data and excluded the possibility of overestimating calibration accuracy. The RMSEs, calculated using the calibration function of *y* = *bx*, are listed in Table 2 along with the RSDs averaged over the six samples. The RMSEs in predicting the concentrations of Mg and Ca based on *I*_Mg_/*I*_Na_ and *I*_Ca_/*I*_Na_ were 1300 mg/kg and 130 mg/kg, respectively. It should be noted from the RSD and RMSE values listed in Table 2 that the intensity normalization was effective at improving precision but not accuracy.

### 3.2. Classification Modeling

The emission peaks of Mg II at 280 nm and Ca II at 393 nm selected for the calibration curves showed significant differences in their intensities across the six salt samples (Figure 5). This indicates that these peaks possess capabilities of discriminating each salt sample from the others. In the following, the two peak intensities were employed as variables to construct a classification model, and the model performance, i.e., classification accuracy, was evaluated based on the *k*-NN algorithm with a separate test data set. The total number of spectra recorded in the experiment is 372 (=62 spectra × 6 salt samples). For the test data set, 12 spectra were randomly selected per each salt sample, and thus 72 spectra (=12 spectra × 6 salt samples) were left for evaluating the classification accuracy. From the other 300 spectra, the (*I*_Mg_, *I*_Ca_) pairs, i.e., objects, were obtained and plotted in Figure 6a. As discussed above, the intensity normalization using *I*_Tot_ or *I*_Na_ was effective at improving measurement precision. The improved precision would decrease the intraclass distribution of objects and separate the six classes further from one another. In Figure 6b,c, the objects, (*I*_Mg_/*I*_Tot_, *I*_Ca_/*I*_Tot_) and (*I*_Mg_/*I*_Na_, *I*_Ca_/*I*_Na_), were plotted, respectively.

The effect of intensity normalization on separating the six classes from one another can be quantitatively evaluated by using the concept of interclass distance, *d*_ij_, defined as below:(3)$$d_{ij}=\frac{\left|m_{i}-m_{j}\right|}{s_{avg}}$$

In the above equation, *i* and *j* indicate two classes between which the interclass distance are measured, and *m_i_* and *m_j_* are the means of intensity or normalized intensity values of the corresponding classes. Thus, |*m_i_* − *m_j_*| is the mean-to-mean distance along a given variable axis. However, the mean-to-mean distance is not sufficient to represent the separation between two data clusters, which have their own variances. It should be scaled by a common standard deviation representing the distribution of objects belonging to the two classes. The pooled standard deviation, *s_pooled_*, expressed below, can be employed for this purpose [26].
(4)$$s_{pooled}=\sqrt{\frac{\left(n_{i}-1\right)s_{i}^{2}+\left(n_{j}-1\right)s_{j}^{2}}{n_{i}+n_{j}-2}}$$

In this equation, *n_i_* and *n_i_* are the numbers of objects belonging to two classes, *i* and *j*, respectively, and *s_i_* and *s_j_* are the standard deviations of the data in the corresponding classes. In this work, the same number of spectra, 50, was used to model each class. In this case, with *n_i_* = *n_j_*, *s_pooled_*^2^ becomes simply an average variance, *s_avg_*^2^ = (*s_i_*^2^ + *s_j_*^2^)/2. Therefore, the interclass distance representing the separation between two classes, *i* and *j*, can be defined as the mean-to-mean distance scaled by *s_avg_*. Figure 7 shows the interclass distances calculated with raw intensities (*I*_Mg_ and *I*_Ca_), those normalized by *I*_Tot_, and those normalized by *I*_Na_. The interclass distance is herein the measure of the two-dimensional space. Thus, those shown in Figure 7 are the two-dimensional Euclidean distances based on the two one-dimensional interclass distances calculated along the two variable axes separately. Figure 7a shows the interclass distances from class A (Australian sea salt) to the others. It indicates that the interclass distances are increased significantly following intensity normalization. Between the two reference signals, *I*_Tot_ and *I*_Na_, the latter is found to be more effective at increasing the interclass distances. This is in good agreement with the results of RSDs shown in Figure 4. Figure 7b–f show interclass distances from each of the classes, C1, C2, K1, K2, and V, to the other classes, respectively. All of these results consistently indicate that the intensity normalization is effective in separating the classes from one another. This would lead to an increase in the classification accuracy. Fifteen interclass distances, *d*_A-C1_, *d*_A-C2_, *d*_A-K1_, *d*_A-K2_, *d*_A-V_, *d*_C1-C2_, *d*_C1-K1_, *d*_C1-K2_, *d*_C1-V_, *d*_C2-K1_, *d*_C2-K2_, *d*_C2-V_, *d*_K1-K2_, *d*_K1-V_, and *d*_K2-V_, were calculated for the three kinds of variables: (i) raw intensities, (ii) those normalized by *I*_Tot_, and (iii) those normalized by *I*_Na_. The average of the 15 interclass distances was 6.0 with raw intensities. This could be increased to 7.0 and 11.9 following normalization using *I*_Tot_ and *I*_Na_ as reference signals, respectively.

The classification accuracy was evaluated for the separate test data set following the *k*-NN algorithm. The three non-parametrically trained models shown in Figure 6 were used to assign the classes of the 72 test spectra. The classes were assigned considering the *k* training objects nearest to the unknown test object. The majority vote criterion was applied with only odd numbers of the *k* nearest training objects. Figure 8a shows the dependence of the classification accuracy obtained by the three models based on (*I*_Mg_, *I*_Ca_), (*I*_Mg_/*I*_Tot_, *I*_Ca_/*I*_Tot_), and (*I*_Mg_/*I*_Na_, *I*_Ca_/*I*_Na_) on *k*, which was varied from 1 to 299. The three models show similar dependence on *k* for classification accuracy. The classification accuracy was maximized at *k* = 1 and 299, and decreased between these values. As shown in the two-dimensional models in Figure 6, most of the objects belonging to different classes are separated, but the objects in classes K1 and V overlap each other. This similarity would be the main cause of the decreasing classification accuracy. In the K1-V overlapping region on the variable space, the majority vote criterion can fail and make the classification accuracy much lower with intermediate numbers of voters than those with *k* = 1 or 299. As discussed above, intensity normalization increased the interclass distances. The model with the larger interclass distances shows a higher classification accuracy over all *k* values and a lesser decrease in classification accuracy at the intermediate *k* values. Finally, the model based on (*I*_Mg_/*I*_Na_, *I*_Ca_/*I*_Na_) with *k* = 1 showing 98.6% classification accuracy was identified as the best one. The corresponding confusion matrix is shown in Figure 8b. Among the 72 test objects, only 1 of them belonging to K1 was misassigned to V.

## 4. Conclusions

A cost-effective LIBS instrument was assembled with a compact low-power DPSSL and a miniature low-resolution spectrometer. In combination with a simple sample preparation method using LPSW substrates, the analytical performances of the LIBS instrument were investigated with six popular edible sea salts consumed in South Korea. Each of the two technologies, the low-cost LIBS instrument and the sample preparation method using LPSW substrates, have been applied for the analysis of edible salts and other materials separately so far. In this work, their combination was applied to the quantification and classification of edible salts for the first time. The concentrations of Mg and Ca could be determined with LODs of 87 mg/kg and 45 mg/kg and accuracies of 0.13 wt% and 0.039 wt%. These analytical capabilities are sufficient for analyzing Mg and Ca in edible salts that typically contain ~1 wt% Mg and ~0.1 wt% Ca. Among the salt samples used in this work, the refined salt (C1) and the sea salt produced in Australia (A) showed very low Mg and Ca contents. It should be mentioned that a higher-performance LIBS instrument or other conventional elemental analysis techniques are required for accurate analysis of Mg and Ca in such low-mineral salts. Also, a classification model showing very high accuracy (98.6% correctness) could be developed using the emission peak intensities of Mg and Ca that are the same as those used for analyzing the concentrations of Mg and Ca. In both quantification and classification, intensity normalization using the Na I emission intensity at 590 nm was effective in improving measurement precision. Herein, formulating the concept of interclass distance, we demonstrated a method to quantitatively estimate the effectiveness of the intensity normalization in increasing discrimination power of a variable used in a classification model. The interclass distances, defined as the scaled mean-to-mean distance, would be useful to measure the effectiveness of any data pre-treatment process at enhancing the classification capabilities of classical models. Finally, it should be emphasized that only a tiny volume (15 μL) of 15 wt% salt solution was needed for quantification and classification based on the measurements of the emission peaks of Mg and Ca. This consumes only 2.25 mg of salt per each sample. Theis analytical capability with a small volume of liquid is potentially useful for analyzing not only water-soluble or -dispensable food materials but also bio-fluids such as urine and blood for medical diagnosis and forensic investigation.

## Figures and Tables

**Figure 1 sensors-23-09280-f001:**
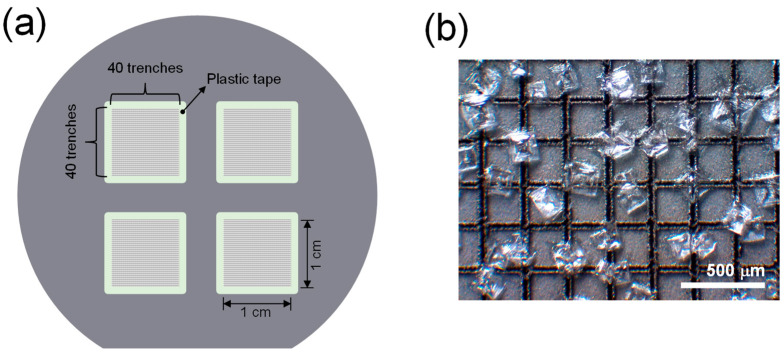
(**a**) Four laser-patterned areas on the silicon wafer and (**b**) the microscope image of the laser-patterned area. In (**b**), the scale bar indicates 500 μm.

**Figure 2 sensors-23-09280-f002:**
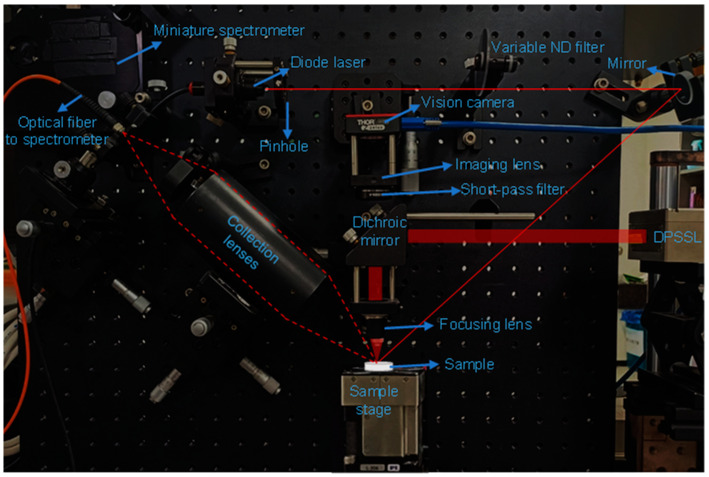
Laboratory-assembled LIBS instrument. The red solid lines indicate the laser beam paths from the diode laser and the DPSSL, and the red dotted line represents the solid angle of the two collection lenses.

**Figure 3 sensors-23-09280-f003:**
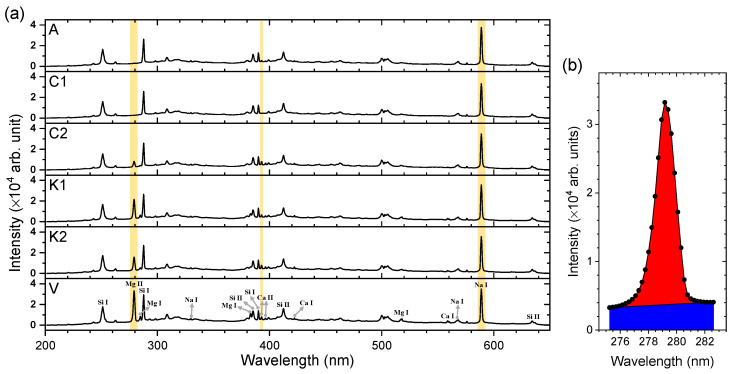
(**a**) Representative LIBS spectra of the samples A, C1, C2, K1, K2, and V. The three emission lines of Mg II, Ca II, and Na I used for analysis are indicated by the yellow bands and (**b**) the expanded spectrum around the Mg II emission peak at 280 nm observed for sample V.

**Figure 4 sensors-23-09280-f004:**
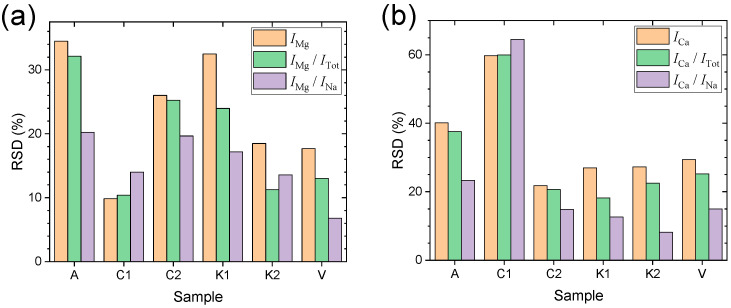
(**a**) RSD values of *I*_Mg_, *I*_Mg_/*I*_Tot_, and *I*_Mg_/*I*_Na_ and (**b**) those of *I*_Ca_, *I*_Ca_/*I*_Tot_, and *I*_Ca_/*I*_Na_.

**Figure 5 sensors-23-09280-f005:**
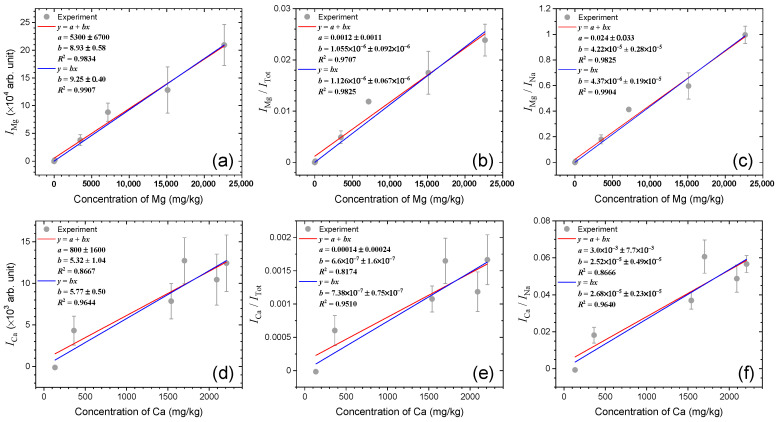
Calibration curves of Mg and Ca constructed using (**a**) *I*_Mg_, (**b**) *I*_Mg_/*I*_Tot_, (**c**) *I*_Mg_/*I*_Na_, (**d**) *I*_Ca_, (**e**) *I*_Ca_/*I*_Tot_, and (**f**) *I*_Ca_/*I*_Na_.

**Figure 6 sensors-23-09280-f006:**
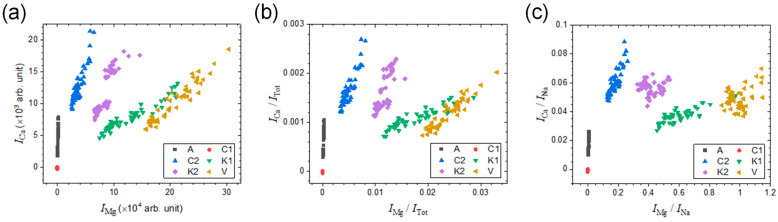
Plots of the objects, (**a**) (*I*_Mg_, *I*_Ca_), (**b**) (*I*_Mg_/*I*_Tot_, *I*_Ca_/*I*_Tot_), and (**c**) (*I*_Mg_/*I*_Na_, *I*_Ca_/*I*_Na_).

**Figure 7 sensors-23-09280-f007:**
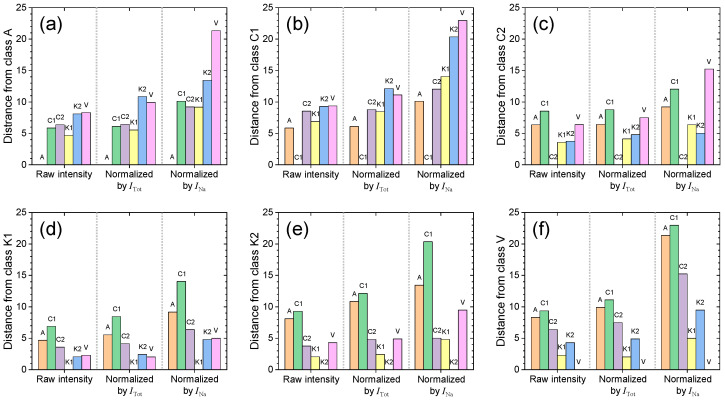
Interclass distances from classes (**a**) A, (**b**) C1, (**c**) C2, (**d**) K1, (**e**) K2, and (**f**) V to the others calculated using raw and normalized intensities.

**Figure 8 sensors-23-09280-f008:**
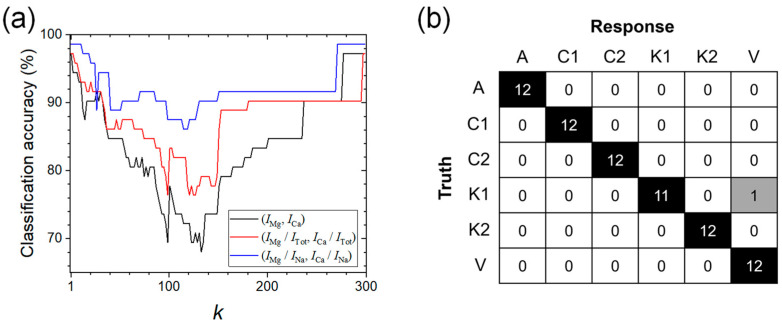
(**a**) Classification accuracy as a function of *k* and (**b**) confusion matrix of the model based on (*I*_Mg_/*I*_Na_, *I*_Ca_/*I*_Na_) at *k* = 1.

**Table 1 sensors-23-09280-t001:** Sample codes, origins, types, and concentrations of Mg and Ca determined by ICP-OES.

Sample Code	Origin	Type	Concentration (ppm)	
			Mg	Ca
A	Australia	Sea salt	75.57 ± 0.17	363.5 ± 4.7
C1	Jiangsu, China	Refined salt	Not detected	134.2 ± 2.4
C2	Shandong, China	Sea salt	3487 ± 10	1701 ± 22
K1	Goheung, South Korea	Sea salt	15,110 ± 230	1540.6 ± 9.5
K2	Sinan, South Korea	Sea salt	7166 ± 95	2210 ± 25
V	Ho Chi Minh, Vietnam	Sea salt	22,650 ± 120	2092 ± 25

**Table 2 sensors-23-09280-t002:** RSDs and RMSEs provided by the raw emission intensities (*I*_Mg_ and *I*_Ca_) and the normalized intensities (*I*_Mg_/*I*_Tot_, *I*_Mg_/*I*_Na_, *I*_Ca_/*I*_Tot_, and *I*_Ca_/*I*_Tot_).

Element	Variable	RSD (%)	RMSE (mg/kg)
Mg	*I* _Mg_	23.1	1300
	*I*_Mg_/*I*_Tot_	19.3	1700
	*I*_Mg_/*I*_Na_	15.2	1300
Ca	*I* _Ca_	34.2	370
	*I*_Ca_/*I*_Tot_	30.7	410
	*I*_Ca_/*I*_Na_	23.1	390

## Data Availability

The data presented in this study are available upon request from the corresponding authors.

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
