# Peer review of "A Simple Laser-Induced Breakdown Spectroscopy Method for Quantification and Classification of Edible Sea Salts Assisted by Surface-Hydrophilicity-Enhanced Silicon Wafer Substrates"

_sensors, 2023, doi:10.3390/s23229280_

Round 1
Reviewer 1 Report
Comments and Suggestions for Authors
This is a well-written article that describes both the methods and results for LIBS analysis of sea salts. The sample preparation method is novel and will be applicable to LIBS analysis of other materials. The statistical methods are sound and well-described. The results show that rapid and reliable identification of provenance is possible. It may be that more elements will need to be used if more sample locations are added to the database, but it is exciting what can be done with Mg, Ca, and Na!
I only have a few comments.
Abstract, line 24. There is an abrupt shift here between giving the LODs for Mg and Ca and the k-nearest neighbors models. It would be useful for the reader to introduce the nearest neighbors work a bit more. For instance, I thought that the paper was going to use more elements to define the nearest neighbors and was surprised to read that this work was done with two elements. Consider something like this: “Peak intensities for Ca and Mg were also used in a k-nearest neighbors model providing….” Just something to say what the model was based on.
In Figure 1, it looks like the salt doesn’t cover the whole substrate. Did some of the laser shots hit the silicon wafer, and if so, how were those shots filtered out?
In Figure 3, the Ca II vertical yellow band is on the Ca 396 peak; the Ca 393 is the leftmost of the two. I might be interesting to label the other peaks for the reader. I see the Ca 422 peak but am not sure of the peak at about 250 nm and the small ones near 500 nm and 640 nm.
Reviewer 2 Report
Comments and Suggestions for Authors
As stated in the title, this study uses Laser-Induced Breakdown Spectroscopy for the elemental quantification and classification of edible salt samples consumed in South Korea. There is no doubt that the used spectroscopic method is versatile and data-rich, and it is vital to gain meaningful insights from the data, especially in the case of food additive analysis.
Generally speaking, a classic methodology to analyze a sample by LIBS and to process spectral data was used without any suggestions to improve what is already known in the literature. Additionally, the same research group has been employing LIBS for the analysis of salt samples over several years, making it challenging to identify the unique aspects of this particular article. For instance, references 24 and 25 employ LIBS for quantifying Ca and Mg in salt samples, while reference 27 introduces the use of LPSW. Moreover, other publications, such as G. Park et al. in 2015 and S. Kumar et al. in 2023, have also explored similar research (low-cost LIBS system for salt analysis). Hence, it would be valuable to emphasize the distinctiveness of the current work.
Listed below are some comments, questions, and suggestions that need to be addressed.
Abstract
- Line 19, Please emphasize the importance of Mg and Ca in salt samples, and why only those elements have been studied?
- Line 20, Please mention the type of used water.
- Line 22, What is the reason for choosing ionic emissions over neutral atomic emission? Do you expect the same results while using neutral atomic emissions?
1. Introduction
- Line 67, “high performances...salts,” is this argument valid for all the minor elements?
- Line 75, In my opinion, the used sample preparation method LPSW, is not the simplification. Can you comment on using other sample preparation methods, e.g., direct deposition on Si wafer without laser patterning, palletization, or directly measuring on salt powders, etc?
- Line 96, In salt analysis, often Na I doublet at 589 nm gets saturated, and then it is not wise to use it for normalization, did you verify this fact? I would also expect the detection and the use of other Na lines, e.g., 330 nm and 616 nm.
2.1 Salt Samples and ICP-OES Analysis
- Line117, as there are two Mg II lines, 279.55 nm and 280. 27 nm, maybe it would be better to specify whole envelop rather than writing only one line at 279.553 nm.
2.2 LIBS analysis
- Figure 1 (b), the crystals are not evenly distributed, is it so that at some locations, salt crystal was measured and at some locations only Si wafer? What is the size of laser crater or focal spot in comparison to salt crystals?
- Line 157, Repetition rate was 1 kHz, and the stage was moving to 1 mm/s. Was it so that there are 1k laser craters in 1mm? And how finally there are 62 spectra from 9.5 cm long scan? Please clarify.
- Figure 2, please give details of ND filter and Vision camera.
3.1 Quantification of Mg and Ca
- Figure 3, What are the acquisition time parameters (delay and exposure time)? What could be the reason of unavailability of Mg I and Ca I lines, as there are neutral atomic lines of Si and Na, present in the figure?
- Figure 3, there is no Mg signal for the samples A and C1, How the intensities were extracted?
- Line 175, Na intensity remain same, is it so that all the samples have same Na concentration?
- How IMg, ICa, and INa, were evaluated, are they peak intensity, integrated intensity, or obtained from fitting of spectral lines?
- Figure 4, both (a) and (b), RSD% for C1 samples, IMg/ INa is higher than others, please justify.
- Line 209, Please provide a reference for equation (1).
- Line 236, Table 2 is missing from the manuscript.
- Line 237, Units for concentrations are not consistent throughout the manuscript, please use one unit among mg/kg, wt.%, or ppm.
- Line 237, are these RMSE values acceptable for the samples of low Mg and Ca concentrations A and C1?
3.2 Classification modeling
- Figure 6, If I understood correctly, the same colour dots represent the intensity values from the 50 measurements of the same samples. What is the reason for such a large dispersion within a cluster. I would expect something like sample C1, for all the samples. Please clarify.
- Figure 7, Please add figure labels (a to f).
Reviewer 3 Report
Comments and Suggestions for Authors
Please see the attachment

Round 2
Reviewer 3 Report
Comments and Suggestions for Authors
Accept in present form
Author Response
The reviewer's comments significantly contributed to the enhancement of our manuscript and the correction of errors. We are deeply grateful for the thorough evaluation of our work. Thank you.